# Development of a Cytotoxic Antibody–Drug Conjugate Targeting Membrane Immunoglobulin E-Positive Cells

**DOI:** 10.3390/ijms241914997

**Published:** 2023-10-08

**Authors:** Aleksandra Rodak, Katharina Stadlbauer, Madhusudhan Reddy Bobbili, Oskar Smrzka, Florian Rüker, Gordana Wozniak Knopp

**Affiliations:** 1Institute of Molecular Biotechnology, Department of Biotechnology, University of Natural Resources and Life Sciences (BOKU), Muthgasse 18, 1190 Vienna, Austria; aleksandra.rodak@boku.ac.at (A.R.); katharina.stadlbauer@boku.ac.at (K.S.); madhusudhan.bobbili@boku.ac.at (M.R.B.); florian.rueker@boku.ac.at (F.R.); 2Ludwig Boltzmann Institute for Traumatology, The Research Centre in Cooperation with AUVA, Donaueschingenstraße 13, 1200 Vienna, Austria; 3Ablevia Biotech GmbH, Maria Jacobi Gasse 1, 1030 Vienna, Austria; oskar.smrzka@ablevia.com

**Keywords:** anti-IgE antibody, antibody–drug conjugate, extracellular membrane proximal domain, IgE-multiple myeloma, internalization

## Abstract

High numbers of membrane immunoglobulin E (IgE)-positive cells are characteristic of allergic conditions, atopic dermatitis, or IgE myeloma. Antibodies targeting the extracellular membrane-proximal domain of the membranous IgE-B-cell receptor (BCR) fragment can be used for specific depletion of IgE-BCR-positive cells. In this study, we derivatized such an antibody with a toxin and developed an antibody–drug conjugate (ADC) that showed strong cytotoxicity for an IgE-positive target cell line. Site-specific conjugation with maleimidocaproyl-valine-citrulline-*p*-aminobenzoyloxycarbonyl-monomethyl-auristatin E via a newly introduced single cysteine residue was used to prepare a compound with a drug–antibody ratio of 2 and favorable biophysical properties. The antibody was rapidly taken up by the target cells, showing almost complete internalization after 4 h of treatment. Its cytotoxic effect was potentiated upon cross-linking mediated by an anti-human IgG F(ab’)_2_ fragment. Because of its fast internalization and strict target specificity, this antibody–drug conjugate presents a valuable starting point for the further development of an anti-IgE cell-depleting agent, operating by the combined action of receptor cross-linking and toxin-mediated cytotoxicity.

## 1. Introduction

IgE class-immunoglobulins (Igs) are present in the blood in very low quantities, yet an acute allergic response can prompt their production by the plasma cells to reach high levels, leading to activation of basophils and mast cells and resulting in consequences ranging in the severity of the symptoms from mild discomfort to anaphylactic shock and death [1]. Current therapies are based on several strategies, such as combating the condition with the use of anti-inflammatory corticosteroids, allergen-specific immunotherapy, or monoclonal antibodies that neutralize the binding of IgE to the high-affinity receptor expressed on the surface of mast cells and basophils [2]. It has turned out that the mechanisms of action of specific anti-IgE biologicals are diverse and strongly dependent on the targeted epitope and the affinity of the antibody [3,4]. Furthermore, specific depletion of IgE-producing cells has been suggested as a viable method to rapidly decrease the level of IgE in the circulation [5,6]. For this purpose, several antibody-based approaches, utilizing the exquisite specificity of their target recognition, have been suggested. The typical target common to these strategies is the extracellular membrane-proximal domain (EMPD, also called CεmX), a part of the B-cell receptor (BCR) of the ε class [7,8,9]. The rationale for targeting this particular domain is based on the observation that the membrane-bound antigen receptor is required for the first signal in B-cell activation; however, without a second signal (support from Th cells and cytokines produced), the B cells are abortively activated and rendered apoptotic or anergic [10,11]. The EMPD region is not present on soluble IgE, which reduces the possibility of antibodies being wasted by on-epitope/off-target cell binding. The antibodies intended for cellular depletion use BCR cross-linking to induce apoptosis or act via the induction of antibody-dependent cellular cytotoxicity [12,13]. The EMPD-targeting antibody quilizumab, exerting both mechanisms and modified by defucosylation to augment the latter activity [14], has been evaluated for the treatment of adults with allergic asthma in clinical phases I and II. It was found to be well tolerated and could indeed decrease total and allergen-specific IgE. However, it did not show a clinically significant impact on adult patients with allergic asthma without standard therapy [15]. 

A further proposed mechanism of action to reduce the activity of IgE-secreting cells is employed by bispecific antibodies, which incorporate the desired target binding together with an effector-engaging moiety and can incite, for example, T-cell activation via the interaction with an activating receptor, such as CD3 [16]. Depending on the target affinity and the actual binding epitopes, such antibodies reach different levels of activation of effector cells, which underlines the importance of both factors for their activity [17]. At the same time, this demonstrates the advantage that can arise from the precise characterization of target recognition properties on the molecular level at the stage of the discovery of targeting agents. Bispecific antibodies can also simultaneously sequester soluble IgE and suppress the differentiation of plasma cells, as well as their IgE synthesis, by co-engaging the inhibitory receptor FcγRIIb and IgE BCR, an activity mediated by the cross-mAb AIMab7195 (formerly XmAb7195), currently in Phase I of clinical trials [18].

Apart from allergic conditions, hyperproduction of IgE is also characteristic of certain types of cancer. High activity of IgE-plasma cells is typical for the aggressive IgE-plasma cell neoplasm [19,20], a very rare cell type of neoplasm, as fewer than 80 cases have been reported [21] since its first mention in 1967 [22]. Compared with malignancies of other immunoglobulin-producing cells, this condition is extremely rare, as it accounts for less than 0.1% of patients; however, its clinical progress is more aggressive [20]. This may be related to its unique clinical features, such as the potent expansion of plasma cells (plasma cell leukemia, PCL) occurring more commonly than in other types of multiple myeloma (MM) [23] and the presence of cytogenetic abnormalities, such as t(11;14) [24]. Due to the high mutational rate, this condition has been classified as one with a high tumor mutational burden (TMB) [25]. Chemotherapy with immunomodulatory imide drugs, proteasome inhibitors, and immunotherapy using anti-CD38 antibodies were not successful [21]. Therefore, potent ablation of such plasma cells would be one of the viable methods to prevent the spread of malignancy into organs.

Although T-cell activating bispecific antibodies have proven efficient in clinical settings, their activity is limited by the count, presence, and activation potential of the patient’s cytotoxic T cells [26]. Additionally, there are several reports of adverse side effects, most commonly cytokine release syndrome, with symptoms of graded severity reaching up to systemic inflammation and death [27]. Another class of antibody-based compounds has gained huge importance in therapy in recent decades: antibody–drug conjugates (ADCs), which combine the fine specificity of antibody targeting with the potent cytotoxic effect of the appended toxin component. In 2022, 14 such therapeutics had reached market approval [28], several of them in the last two years [29,30]. More than 100 candidate molecules are currently in advanced stages of clinical development. Until now, all of them were intended for cancer therapy, including both blood and solid tumors. 

In this study, we set out to derivatize an EMPD-targeting antibody with a toxic moiety to potentiate its effect on Fcε-expressing cells. The antibody 15cl12 [31] cannot bind to soluble IgE, which is advantageous as its concentration in pre-malignant IgE-monoclonal gammopathy of undetermined significance (MGUS) conditions as well as in IgE-MM can rise several thousand times above normal values, up to several mg/mL [32,33]. This elevated IgE level can deplete the therapeutic agent intended to address IgE-producing cells. As a naked antibody, 15cl12 could dose-dependently inhibit the proliferation of the cell lines, stably expressing the Fcε fragment, including EMPD, in vitro [31]. The inhibitory activity is not limited to EMPD-targeting antibodies; it is also caused by a cross-link via multivalent binding to the Cε3 domain, mediated by the clinically approved antibody omalizumab. Furthermore, when used in conjunction with 15cl12, the pro-apoptotic activity of omalizumab was further enhanced. When applied for passive immunization, 15cl12 significantly suppressed the induced total and antigen-specific IgE levels in a mouse model expressing the human long EMPD sequence instead of the mouse variant [31]. We further characterized this antibody and its activity as an ADC. Among several options available for toxin coupling, site-specific conjugation utilizing lone cysteine residues is attractive as it can deliver a homogeneous antibody–drug preparation [34]. Further, the location of the toxin moiety influences the manufacturability as well as the conjugate stability and efficacy in vivo [35]. Therefore, we have chosen to conjugate the toxin to the newly introduced cysteine residue introduced at position L328 of human IgG1 (EU numbering [36]), regarding the favorable physicochemical and pharmacokinetic properties of such antibody conjugates described previously [37,38]. 

## 2. Results

### 2.1. Antibody Production and Characterization

15cl12 was expressed as a chimeric antibody with human constant domains (all sequences in Appendix A) and was purified from the supernatant of HEK293-6E cells at about 30 mg/L. For site-specific coupling of the toxin, the L328C variant of this antibody was used (Figure 1a), and this single amino acid mutation had no negative impact on its expression yield. Coupling of maleimidocaproyl-valine-citrulline-*p*-aminobenzoyloxycarbonyl-monomethyl-auristatin E (MC-Val-Cit-PAB-MMAE) proceeded after mild reduction and reoxidation of the antibody [37,39]. Parental antibodies, cysteine-containing scaffold variants, and the coupled antibody all showed a single band at the expected size in SDS-PAGE analysis (Figure 1b). Size-exclusion chromatography (SEC) performed in native conditions showed a single sharp monomeric peak for the conjugate, free of aggregates and degradation fragments (Figure 1b). Hydrophobic interaction chromatography (HIC) analysis of the conjugated antibody revealed a single predominant species with a longer retention time compared to the parental antibody, indicating an increase in hydrophobicity (Figure 1b). Mass spectrometry analysis delivered evidence of unmodified antibody light chains and heavy chains linked to one molecule of toxin, with a difference in mass of 1319.5 Da from the uncoupled to the coupled variant (compared with 1316.63 Da expected) (Figure 1c).

### 2.2. Binding Properties of 15cl12 and the ADC Derivate

The antigen affinity of the antibodies was first compared using a biolayer interferometry experiment (BLI) with a synthetic peptide ligand representing a section of the human EMPD [31]. The measured affinity amounted to 11 nM (steady state value) for the unmodified antibody and 7.5 nM for the toxin-armed variant (Figure 2a). ELISA with the same peptide used for coating, parental, cysteine-modified, and conjugated antibodies showed similar reactivity with an EC_50_ of 0.23 nM (Figure 2b). To examine cellular binding, Ramos cells were stably transformed with an EMPD-containing Fcε construct (Edox), and a control cell line (Cdox) [31] was induced with doxycycline for 48 h. The toxin-coupled antibody bound to Edox cells with an EC_50_ of about 8 nM, which was higher than the 4 nM determined for the parental antibody (Figure 2c). These values were consistent through the experiments, within repetitive measurements with different preparations of both molecules and even when a different secondary antibody, namely an anti-human gamma chain conjugate, was used for detection (Appendix A). There was no reactivity with the control cell line except for a marginal signal at the highest concentration point of 150 nM for both molecules.

### 2.3. Internalization Experiments

We first assessed the internalization properties with the N-hydroxysuccinimide (NHS)-Alexa Fluor 488-coupled parental 15cl12 antibody. We compared the fluorescence of stained Edox cells incubated with or without the fluorescence-quenching antibody. When quenching is performed, only the signal from the internalized antibody can be measured. When the incubation with the labeled 15cl12 proceeded on ice, about 15% of the fluorescence could not be quenched anymore (Figure 3a). When the incubation was performed in conditions that favor internalization in a time interval from 0.5 to 4 h at 37 °C, after 4 h, almost 90% of the fluorescence was not quenched, indicating efficient internalization of the antibody. Alexa Fluor-labeled non-binding antibody cetuximab was used as a negative control and indeed showed no reactivity. Microscopic observation of the cellular surface confirmed the strong surface staining of Edox cells when stained on ice and potent internalization at 37 °C, as well as the specificity of the 15cl12 antibody, as there was no reactivity with the control antigen-negative Cdox cell line (Figure 3b).

### 2.4. Comparison of Binding and Internalization Properties with Other Anti-IgE Antibodies

We compared 15cl12 with other IgE-specific antibodies that have already entered clinical trials. Omalizumab (Xolair^®^) and ligelizumab both target the Cε3 domain, although at different epitopes [3]. Omalizumab is approved for the treatment of moderate to severe asthma, and ligelizumab showed even superior functionality to omalizumab in the therapy of chronic spontaneous urticaria (CSU) in Phase IIb but not in Phase III [40]. On the other hand, quilizumab binds to the EMPD region of membranous IgE. The antibodies were expressed and purified as described above, and their monomeric profile was monitored with SEC-HPLC in native conditions (Appendix A). All antibodies could react with the IgE-expressing cell line (Figure 4a), but the binding of quilizumab was weaker than with 15cl12 and others. We have also examined if quilizumab and 15cl12 compete for the same epitope, but no competing effect for the cell surface-bound antigen could be determined (Appendix A). The EC_50_ of cell surface binding of omalizumab and ligelizumab was notably lower than that of 15cl12 at about 0.1 nM. We then determined the effect of antibodies on the internalization of Fcε. We found that upon incubation at 37 °C, the surface fluorescence of Edox cells decreased by about 70% when they were treated with omalizumab or quilizumab and by over 80% when they were treated with 15cl12 or ligelizumab in comparison with the incubation on ice (Figure 4b). The temperature of incubation caused no difference in fluorescence when Edox cells were incubated without any antibodies.

### 2.5. Specific Cytotoxic Activity of the ADC

First, we evaluated the direct cytotoxic activity of the antibodies upon their addition to Edox cells. Toxin-conjugated and unconjugated non-binding antibody trastuzumab was also included as a negative control. There was no effect on the unconjugated cysteine-modified antibody and trastuzumab variants on any of the cell lines, and the control Cdox cells were also not affected by the toxin-coupled 15cl12. However, the conjugate reduced the viability of the target-positive cell line dose-dependently with an EC_50_ between 10 and 100 nM (Figure 5a). The cytotoxic effect could be potentiated by applying additional cross-linking with a (Fab’)_2_ fragment. When the cross-linking (Fab’)_2_ fragment was added at the same molar concentration as the primary antibody, the 15cl12-toxin conjugate reduced the viability of the target cells significantly even when applied at a low concentration of 0.6 nM (considering only two toxin payloads per antibody molecule) (Figure 5b and Appendix A). However, the uncoupled antibody had no effect, and the viability of the control cell line was not affected by either of the antibody variants at 10 nM or below.


## 3. Discussion

We have prepared a toxin-coupled antibody targeting the EMPD of cell membrane-bound IgE. The antibody bound specifically to the target cell line with high affinity and showed strong internalization. For toxin coupling, we have chosen site-specific labeling at the unique newly introduced cysteine residue, and indeed, mass spectrometry in reducing conditions confirmed a homogenous drug-to-antibody ratio of two by a shift in molecular weight that fits one toxin molecule per heavy chain. The novel ADC was further tested for activity on antigen-positive cell lines and could potently inhibit proliferation of the Fcε-expressing cells, while it did not affect the control line, and there was no toxicity for any of the cell lines by non-binding antibody-derived conjugates prepared with the same protocol. The EC_50_ of inhibition between 10 and 100 nM could be lowered by about 20-fold when additional F(ab’)_2_-mediated cross-linking was used. The cross-linking of EMPD by the antibody itself leads to an antiproliferative effect [31]. While the toxin-coupled antibody was more cytotoxic to antigen-positive cells than the naked antibody, additional cross-linking with the Fab enhanced this effect. The final EC_50_ values reached compare well with the recently developed MMAE-conjugated ADCs [41]. Although five MMAE-conjugated antibodies have been approved for human use [42], recent advances in the design of toxic payloads with increased potency are rapidly expanding the spectrum of options available for antibody conjugation. These can be tailored to the particular antibody and its intended use, providing not only increased potency but also improved safety and efficacy by overcoming resistance mechanisms [43].

Comparing our results with the report describing the original variant of 15cl12 [31], we expected that the chimeric 15cl12 as a naked antibody would notably reduce the viability of the target Edox cell line. When expressed with human constant domains, the affinity to the model peptide representing a part of EMPD was still very high at 11 nM, and the antibody specifically and potently reacted with the target cells. However, it is known that the 5-ethynyl 2′-deoxyuridine (EdU)-incorporation method, originally used for the estimation of the antiproliferative effect of 15cl12, is by far more sensitive in detecting growth-inhibiting effects than the methods that directly address cellular viability, such as 3-(4,5-dimethylthiazol-2-yl)-2,5-diphenyltetrazolium bromide (MTT) incorporation [44]. The ATP-production assay used here also detects the response of the cells that might not be able to enter cell division but are still metabolically active and cannot be discriminated from proliferating cells [45]. Nevertheless, this method is recognized as a valuable and commonly used tool in the discovery and characterization of recent ADC candidates [46,47].

The efficient internalization of the 15cl12 antibody is an important property that justifies its derivatization into an ADC, as shown by monitoring the time course study of internalization, which was confirmed by the absence of a response from an irrelevant labeled non-binding antibody (cetuximab) labeled in the same way. Although there are reports of non-internalizing ADCs that performed well in animal models [48], it is generally believed that fast cellular uptake is a pre-requisite of ADC cleavage to liberate the active toxin component, contributing to a lower degree of peripheral lysis [49]. Moreover, it has already been shown that more rapid internalization favors the activity of immunotoxin conjugates [50] and ADCs [51]. It is likely that the biological activity of 15cl12 could be further improved by affinity maturation, especially by extending its off-rate, as this has been suggested to be a critical factor for the antiproliferative effect among EMPD-targeting antibodies [31].

After first demonstrating the potential of 15cl12 to be developed into an ADC in this study, we were also interested in the higher degree of cross-linking imposed via this antibody, which can lead to a more potent biological effect. Indeed, the 15cl12-based ADC was much more deleterious to the target cells when additional cross-linking via anti-human IgG (Fab’)_2_ was present. However, the translational value of this observation will probably not be harvested in an application of ADC in conjunction with a crosslinker, as the agents requiring such a regimen are disfavorable in comparison with single antibodies with the same effect, as described for CD40-targeting candidates [52]. Potentiation of the cytotoxic effect could instead be realized by the introduction of additional antigen binding sites within one antibody, either homoparatopic or alternatively targeting other Fcε domains but EMPD. Recently reported IgE-targeting alternative binding scaffolds, which are smaller than antibodies and of excellent biophysical properties [53,54], would also be well suited as components of such bi- or multispecific agents.

On the other hand, the more potent cytotoxic activity of 15cl12 upon cross-linking could be harvested for a pretargeting application strategy. In such dosing regimens, the target-specific antibody is first applied, and in a second step, a molecule with a short half-life in vivo, conjugated with a toxin of a radioactive compound, is introduced. Proof-of-concept preclinical and clinical studies have shown that such methods can substantially increase the therapeutic index [55].

To conclude, the 15cl12 antibody has the potential to be developed into an ADC for managing pathologic conditions with increased numbers of cells expressing membrane-bound IgE. The most important improvement required is the affinity increase to the cell-presented antigen, and the most important validation is the confirmation of in vitro toxicity results with in vivo testing using a human EMPD mouse model [31].

## 4. Materials and Methods

### 4.1. Protein Expression, Purification, and Preparation of Toxin Conjugates

Antibody sequences were cloned into pTT5-based vectors (Canadian National Research Council (CNRC) Ottawa, Ontario, Canada) and introduced into suspension HEK293-6E cells (CNRC) using polyethylenimine (PEI)-mediated transfection with 1 µg DNA and 2 µg PEI per mL cell suspension, with antibody heavy to light chain at 1:1 mass ratio. Cells were kept in an F17 medium with 4 mM glutamine, 0.1% Pluronic F-68, and 0.25 µg/mL G-418 (all Fisher Scientific, Hampton, NH, USA) on an orbital shaker at 125 rpm at 37 °C in a humidified atmosphere with 5% CO_2_ for 5 days, with an addition of 0.5% TN-1 (Organotechnie, La Courneuve, France) on day 2 post-transfection. The supernatant was harvested by centrifugation at 2300× *g* for 20 min at 4 °C and filtered through a 0.45 µm filter. After buffering to 0.1 M sodium phosphate, pH 7.0, it was loaded onto a HiTrap 1 mL Protein A column (Cytiva, Marlborough, MA, USA) used in combination with an ÄKTA purifier system (Cytiva) at 1 mL/min, and loading was completed with the same buffer. Antibodies were eluted in 1 mL fractions using 0.1 M glycine, pH 3.5, which were neutralized immediately with the addition of 6 μL of 2 M Tris. Protein concentration was determined with Nanodrop (Fisher Scientific), and fractions of interest were dialyzed in Snakeskin tubing with a molecular weight cut-off (MWCO) of 10,000 Da (Fisher Scientific) against at least a 100-fold volume of phosphate-buffered saline (PBS) (Fisher Scientific) overnight at 4 °C. The concentration of antibodies was remeasured, and preparations were stored at −80 °C until further use.

15cl12 antibody solution at 1 mg/mL was incubated with 250 µM Tris (2-carboxyethyl) phosphine hydrochloride (TCEP) (Sigma-Aldrich, St. Louis, MO, USA) for 2 h at 37 °C, and buffer was then exchanged to PBS/1 mM EDTA, pH 7.4, using Zeba^TM^Spin columns with 40 kDa MWCO (Fisher Scientific) according to the manufacturer’s instructions. Quenching with 133.3 µM dehydroascorbic acid (DHAA) (Sigma-Aldrich) was performed at 25 °C for 1.5 h, and the protein preparation was incubated with an 8-fold molar excess of MC-VC-PAB-MMAE (ALB Technologies, Henderson, NV, USA) with a molecular weight of 1316.63 Da for 2 h at 25 °C. Buffer was exchanged to PBS using Slide-a-Lyzer dialysis cassettes (MWCO 10 kDa) (Fisher Scientific) at 4 °C overnight. 

### 4.2. Protein Characterization

#### 4.2.1. SDS-PAGE

Two µg of antibody preparations were mixed with loading sample buffer and applied to 4–12% Novex NuPAGE gels, together with the Mark12 Unstained molecular weight marker. Electrophoresis was run in MES buffer for 35 min at 200 V. Gels were fixed in 50% ethanol and 10% acetic acid for 30 min, stained with the Novex Colloidal Blue staining kit (all chemicals from Fisher Scientific) overnight, and de-stained with distilled water overnight.

#### 4.2.2. SEC-HPLC

A total of 20 µg protein at about 1 mg/mL were loaded on a Superdex 200 Increase 10/300 GL column (Cytiva) connected to a Shimadzu LC-20A Prominence system equipped with a diode array detector. Chromatography runs were performed in PBS with 200 mM NaCl, pH 7.0, at a constant flow rate of 0.75 mL/min. Molecular weight standards ranging from 670 to 1.3 kDa (BIO-RAD, Hercules, CA, USA) were used for column calibration. 

#### 4.2.3. HIC

All solutions were prepared using water purified via a Milli-Q water purification system (Millipore, Burlington, MA, USA) and filtered through a 0.1 μm PVDF filter before use. HIC analysis was performed using an LC-20A Prominence system (Shimadzu, Kyoto, Japan) equipped with a binary/quaternary pump, PDA detector, column thermostat, degasser, and temperature-controlled autosampler, with the autosampler cooled to 4–8 °C and the column oven temperature set to 25 °C. Ten µg protein were injected into a MAbPac HIC Butyl 5 µm, 7.8 × 100 mm (Sepax Technologies, Newark, DE, USA), and the chromatography was run at a flow rate of 1 mL/min. The equilibration was in 100% mobile phase A (25 mM Tris.HCl, pH 7.5, 1.5 M ammonium sulfate), and following sample injection in 2.5 mL of 100% mobile phase A (25 mM Tris.HCl, pH 7.5, 1.5 M ammonium sulfate), the proteins were eluted in a linear gradient from 0% B to 100% B (25 mM Tris.HCl, pH 7.5, 20% (*v*/*v*) isopropanol) in 15 mL. The column was regenerated with 5 mL of 100% B, re-equilibrated with a gradient from 0% A to 100% A in 10 mL, and rinsed with 21.5 mL of 100% A. 

#### 4.2.4. Mass Spectrometry Analysis

The pH of the samples was set to 8.0 with the addition of 100 mM Tris. HCl and digestion with PNGase (Roche, Basel, Switzerland) were followed overnight at 37 °C for deglycosylation. Reduction was performed with 1.5 mM dithiothreitol (DTT) at 37 °C for 45 min, and samples were then analyzed immediately. Two µg of the samples were directly injected into an LC–ESI–MS system (LC: 1290 Infinity II UPLC, Agilent Technologies, Santa Clara, CA, USA). A gradient from 15 to 80% acetonitrile in 0.1% formic acid (using a BioResolve column (2.1 × 5 mm), Waters, Milford, MA, USA) at a flow rate of 400 μL/min was applied (9 min gradient time). Detection was performed with a Q-TOF instrument (6230B LC-TOFMS, Agilent Technologies) equipped with the Jetstream ESI source in positive ion, MS mode (range: 100–3200 Da). Instrument calibration was performed using an ESI calibration mixture (Agilent Technologies). Data were processed using MassHunter BioConfirm B.08.00 (Agilent Technologies), and the spectrum was deconvoluted with the MaxEnt algorithm.

### 4.3. Target Antigen Binding

#### 4.3.1. Biolayer Interferometry

BLI experiments were performed with an Octet^®^RED 96e apparatus (ForteBio, Sartorius, Göttingen, Germany). Biotinylated EMPD-characteristic peptide (GGSAQSQRAPDRVLCHSGQQQGLPRAAGGSVP-K-biotin) was purchased from Fisher Scientific and used for coating of streptavidin sensors (ForteBio, Sartorius) at 5 µg/mL. Antibodies in two-fold dilutions in assay buffer (PBS with 1× Kinetics buffer, ForteBio, Sartorius) starting from 250 nM were allowed to associate for 15 min and dissociate for 20 min into the same buffer at 25 °C with shaking at 1000 rpm. After subtracting the response of antibodies binding to an uncoated tip and the baseline recorded as a response of a peptide-coated tip immersed in assay buffer, the data were analyzed with the ForteBio Data Analysis package, Version 11.0, and a two-to-one binding model was used for the fitting of raw data.

#### 4.3.2. ELISA

Biotinylated EMPD-characteristic peptide was immobilized to the wells of a 96-well Streptavidin-Immobilizer plate (NUNC, Fisher Scientific) at 30 nM (0.11 µg/mL) in 100 μL PBS per well for 1 h at room temperature (RT). After washing three times with 200 µL PBS, plates were blocked with 200 µL 4% bovine serum albumin (BSA)–PBS for 1 h at RT, and antibodies were added in 100 µL volume in 3-fold dilutions in 2% BSA–PBS starting from 45 nM. After a 1 h incubation and three washes with 200 µL PBS, anti-human kappa chain-horseradish peroxidase conjugate (A-7164, Sigma-Aldrich) was added in a 1:5000 dilution in 2% BSA–PBS for 45 min at RT. Bound antibody was measured after the addition of 100 µL 3,3’, 5,5′-tetramethylbenzidine (TMB) substrate (Sigma-Aldrich) and stopping of the reaction with an equal volume of 30% H_2_SO_4_ at 450/620 nm (Spark, Tecan, Männedorf, Switzerland). EC_50_ of binding was determined using GraphPad Prism Version 5.00 (GraphPad Software, San Diego, CA, USA).

#### 4.3.3. Cell Culture

An EMPD-expressing cell line, denoted as the Edox cell line (kind gift of Oskar Smrzka and Günther Staffler, AFFiRiS AG, Vienna, Austria), was established by stably transforming the Ramos cell line with an all-in-one TET-inducible lentiviral HIV-based construct. This construct encodes the IgE-Fc-B-cell receptor (BCR) encompassing the 3×FLAG tag-Cε2-Cε3-Cε4-EMPD-transmembrane-intracellular domain. Additionally, Ramos cells transformed with an empty vector (Cdox cell line) were cultivated in RPMI-1640 with 2 mM L-glutamine, sodium pyruvate, 100 U/mL penicillin, and 100 μg/mL streptomycin with 0.3 µg/mL G-418 (all from Fisher Scientific) and 10% fetal calf serum (FCS) (Sigma-Aldrich) (complete RPMI medium) at 37 °C under 5% CO_2_ in a humidified atmosphere. Cell surface expression of IgE-BCR was induced with 1 µg/mL doxycycline (Clontech, Mountain View, CA, USA) for 48 h before the experiment.

#### 4.3.4. Cell Staining

Cell number and viability were determined with the TC20 Automated Cell Counter (BIO-RAD) using the Trypan blue exclusion method. Cells were harvested with centrifugation at 300× *g* for 5 min at 4 °C, resuspended in 2% ice-cold BSA–PBS at a density of 2 × 10^6^ cells/mL, blocked for 30 min on ice, and transferred into a 96-U-shaped-well plate in 100 µL aliquots. All stainings of Edox cells were performed in duplicates, and a single staining was conducted for Cdox; two independent experiments were performed. After centrifugation at 300× *g* for 5 min at 4 °C, the blocking solution was removed, and cell pellets were resuspended in 100 µL of antibody solution in 2% BSA–PBS in graded concentrations in 2.5-fold steps, starting at 150 nM. After a 30-min incubation on ice, cells were collected with another centrifugation step and incubated in 100 µL/well of anti-human kappa light chain-fluorescein isothiocyanate (FITC) (RRID: AB_259557; F-3761, Sigma-Aldrich), diluted 1:100 in 2% BSA–PBS and kept on ice, protected from light. After a final centrifugation step, cells were resuspended in 200 µL ice-cold PBS, and 10,000 cells per sample were analyzed with a Guava^®^ easyCyte™ Flow Cytometer (Luminex, Austin, TX, USA). Kaluza Analysis 2.1 software (Beckman Coulter, Brea, CA, USA) was used to determine the geometric mean of cell fluorescence, and EC_50_ of cell surface binding was determined using Prism5 software (GraphPad). As an alternative to the anti-human kappa light chain conjugate, an anti-human IgG (γ-chain-specific) F(ab′)_2_ fragment conjugated with R-phycoerythrin (PE) (RRID: AB_261189, P-8047, Sigma Aldrich), diluted 1:1000, was used. For cell staining with omalizumab, ligelizumab, and quilizumab, the same protocol was followed, only that 3-fold dilutions starting at 30, 4.5, and 300 nM were used, respectively.

#### 4.3.5. Competition between EMPD-Binding Antibodies 15cl12 and Quilizumab

Edox cells were blocked as described before and then incubated with 100 µL of 100 nM quilizumab, 100 nM non-binding control antibody trastuzumab, or 50 nM 15cl12, diluted in 2% BSA–PBS for 30 min on ice, and a parallel sample did not contain any antibody (non-treated cells). Cells were then centrifuged at 300× *g* for 5 min at 4 °C and resuspended in 100 µL of 15cl12-Alexa Fluor 488 conjugate, diluted to 50 nM concentration in 2% BSA–PBS. After 30-min incubation on ice and another centrifugation step, cells were resuspended in 200 µL ice-cold PBS, and the fluorescence was measured with a Guava^®^ easyCyte™ Flow Cytometer (Luminex). Percent fluorescence was determined as (mean fluorescence intensity (MFI) of non-treated cells–MFI of treated cells)/MFI of non-treated cells × 100. All samples were stained in triplicate. 

### 4.4. Internalization Assays

#### 4.4.1. Quenching of Surface Fluorescence

The 15cl12 antibody was labeled using the Alexa Fluor 488 protein labeling kit (A10235, Fisher Scientific) with a molar ratio of 10:1 (reagent to antibody), otherwise following the manufacturer’s instructions. Cetuximab conjugate was prepared in the same way and was used as a negative non-binding control. 

Induced Edox cells were incubated with 50 nM antibodies coupled with NHS-Alexa Fluor 488 in complete RPMI medium at 200,000 cells/well, on ice, or at 37 °C for 0.5, 2 h, or 4 h in triplicate. Samples were then placed on ice for 5 min, the antibody solution was washed away, and each sample was resuspended in 100 µL 2% BSA–PBS with or without 50 µg/mL anti-Alexa Fluor 488 quenching antibody (RRID: AB 221544, A11094, Fisher Scientific) for 30 min on ice. Cells were pelleted at 300× *g* for 5 min at 4 °C, resuspended in 200 µL of ice-cold PBS, and analyzed with a Guava^®^ easyCyte™ Flow Cytometer (Luminex). Percent quenched fluorescence was calculated as: (1-mean fluorescence intensity (MFI) (quenched sample)/MFI (non-quenched sample) × 100.

#### 4.4.2. Microscopy

For internalization, induced Edox and Cdox cells at a density of 1 × 10^6^ cells/mL were incubated with a 50 nM solution of antibodies coupled with NHS-Alexa Fluor 488 in a complete RPMI medium at 37 °C overnight. Cells were harvested with centrifugation at 300× *g* for 5 min at 4 °C, washed twice with 1 mL PBS at RT, and stained with Hoechst 33,342 (BIO-RAD), diluted to 1 µg/mL in PBS for 5 min at RT. Staining was completed by rinsing cells twice with 1 mL of ice-cold PBS. For the measurement, cells were resuspended in 400 µL ice-cold PBS, and 10 µL of cell suspension were delivered into a slot of a 4-well microinsert in 35 µm µ-dish (Ibidi, Gräfelfing, Germany). Samples were analyzed with a DMI6000B microscope (Leica, Wetzlar, Germany) using HCX Objective Plan-Apochromat 63×/1.4 Oil. Data on sample fluorescence were processed using Leica Application Suite X software, version 3.7.0.

The control experiment to monitor cell surface staining proceeded in a similar way, except for antibody incubation, which was for 30 min on ice. 

#### 4.4.3. Internalization of Fcε

Omalizumab, ligelizumab, and quilizumab were prepared as IgG1-format antibodies, as described above. Edox cells were harvested and incubated with antibodies at the concentration corresponding to saturation concentration for staining (10 nM for 15cl12, 3.33 nM for omalizumab, 1.5 nM for ligelizumab, and 100 nM for quilizumab), in RPMI with 10% FCS and 2 mM L-glutamine, sodium pyruvate, 100 U/mL penicillin, and 100 μg/mL streptomycin, at 200,000 cells/well, on ice, and at 37 °C for 4 h. Samples were then placed on ice for 5 min, resuspended in 10 µg/mL anti-FLAG antibody (RRID: AB_439712, F-3040, Sigma-Aldrich) in 2% BSA–PBS, and incubated on ice for 30 min. The binding of anti-FLAG was detected with goat anti-mouse-FITC conjugate (AB_259490, F-2653, Sigma-Aldrich), diluted 1:200 in 2% BSA–PBS, after 30 min incubation on ice. Cells were resuspended in 200 µL ice-cold PBS and analyzed with a Guava^®^ easyCyte™ Flow Cytometer (Luminex). A decrease in fluorescent signal upon treatment at 37 °C was calculated as (1-(MFI at 37 °C/MFI on ice)) × 100.

### 4.5. Cytotoxicity Experiments

#### 4.5.1. Direct Specific Cytotoxic Activity of the ADC

Induced antigen-positive and control cells were seeded at a density of 5000 cells per well into the wells of a 384-well plate in a 12.5 µL complete RPMI medium. Another 12.5 µL volume containing an aliquot of a three-fold serial dilution of antibodies in complete RPMI, starting at 400 nM (200 nM final concentration), was added. The plate was sealed with a semi-permeable membrane (BreathEasy, Sigma Aldrich), and cells were incubated at 37 °C in a humidified atmosphere with 5% CO_2_ for 8 days. Proliferation was measured using the CellTiter-Glo^®^ 2.0 substrate (Promega, Madison, WI, USA). Plates were equilibrated to RT, and 25 µL substrate per well were added. Luminescence was measured immediately using a Spark plate reader (Tecan). The luminescence in wells with untreated cells was considered to correspond to 100% viability, while the luminescence in wells without cells corresponded to the background. Percent viability was calculated as (luminescence in test well/luminescence in wells with untreated cells) × 100 after the background luminescence has been subtracted from all readings. Measurements were performed in triplicate, and the results were processed using GraphPad Prism 5.

#### 4.5.2. Cytotoxicity after Cross-Linking of Anti-IgE Antibodies

For cross-linking, goat F(ab’)_2_ fragment to human IgG (RRID: AB_2334288, 55049, Cappel, Malvern, PA, USA) was used. After reconstitution of the reagent, preparative size exclusion chromatography on a *Superdex* 200 Increase 10/300 GL column was performed as described above to isolate the monomeric protein of about 100 kDa in size (Appendix A). Cells were dispensed at a density of 200,000/well on a 24-well plate in a 0.25 mL complete medium. To each well, 0.25 mL of 4-fold serial dilutions of antibodies starting at 20 nM (end concentration 10 nM) was added for 1 h, then removed using centrifugation at 300× *g* for 5 min at RT and replaced with a solution of the crosslinker with the same molar concentration. Plates were sealed with a semi-permeable membrane and then incubated and processed as described above. Quadruplicate parallel samples were examined, and after data evaluation, the statistical significance of the results was calculated using one-way ANOVA in Graph Pad Prism 8.0.2. 

## Figures and Tables

**Figure 1 ijms-24-14997-f001:**
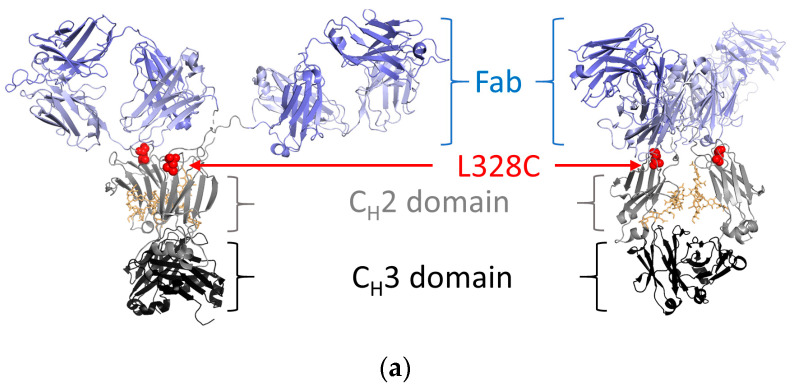
Design and characterization of 15cl12 as an antibody–drug conjugate. (**a**) Location of the L328C mutation (the figure was prepared with PyMol, version 2.4.0 (Schrödinger Inc., New York, NY, USA), using PDB: 1HZH); (**b**) Results of SDS-PAGE analysis, M: Mark 12 unstained marker (left), size exclusion chromatography in native conditions (center) and hydrophobic interaction chromatography (right), MWS: molecular weight standard containing proteins of 670, 158, 44, 17, and 1.3 kDa in size, AU: arbitrary units; (**c**) mass spectrometry analysis of reduced unconjugated and conjugated antibody. 15cl12_L328C*tox: 15cl12_L328 coupled with toxin.

**Figure 2 ijms-24-14997-f002:**
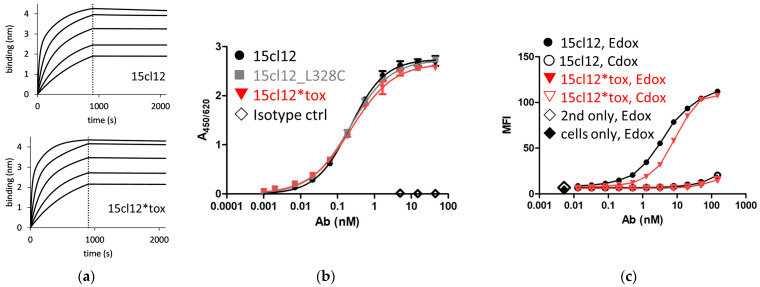
Antigen binding properties of 15cl12 and the toxin derivate. (**a**) Results of a biolayer interferometry experiment with extracellular membrane-proximal domain (EMPD)-characteristic peptide as a ligand (binding in nm, nanometer); (**b**) ELISA with EMPD-characteristic peptide as a coating; (**c**) results of a flow cytometry experiment showing binding to antigen-positive (Edox) cells and antigen-negative (Cdox) cells. 15cl12*tox: 15cl12_L328 coupled with toxin; ctrl: control; second only: secondary reagent only.

**Figure 3 ijms-24-14997-f003:**
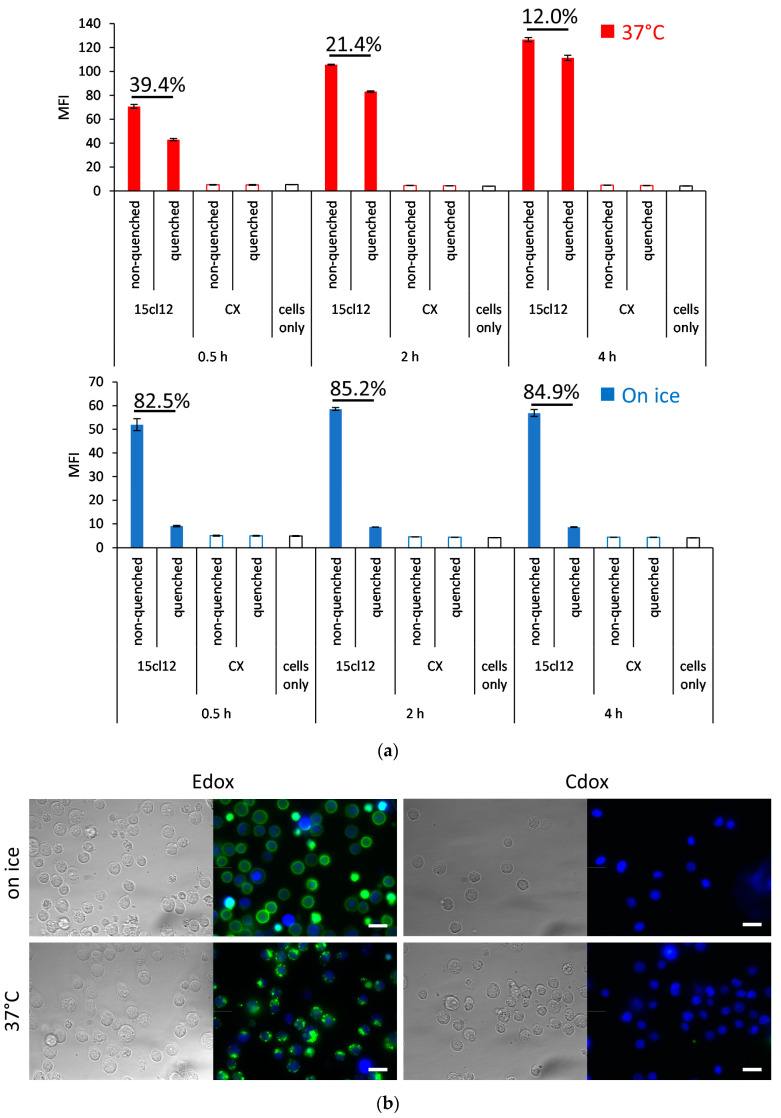
Internalization of 15cl12. (**a**) Internalization analysis using quenching of surface fluorescence with an Alexa Fluor 488 quenching antibody after treatment with 15cl12 or non-binding control antibody cetuximab (CX) for 0.5, 2, and 4 h at 37 °C or on ice. Percent quenched fluorescence calculated as (1-mean fluorescence intensity (MFI) (quenched sample)/MFI (non-quenched sample) × 100 is indicated; (**b**) Fluorescent microscopy showing bright field image, 15cl12 antibody staining (green), and Hoechst 33,342 staining (blue) of Edox and Cdox cell lines after incubation on ice or at 37 °C, scale bar = 20 μm.

**Figure 4 ijms-24-14997-f004:**
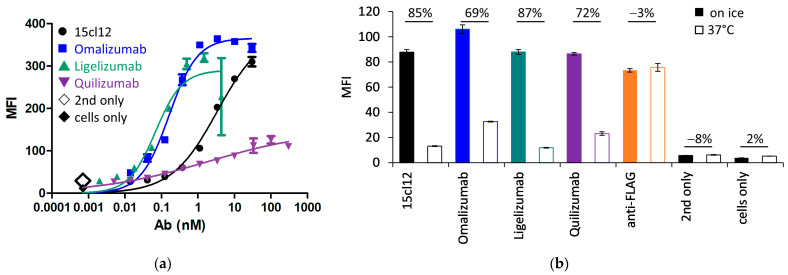
Cell binding and internalization of Fcε of 15cl12 were compared with omalizumab, ligelizumab, and quilizumab. (**a**) Binding of antibodies to antigen-positive (Edox) cells. MFI: mean fluorescence intensity; (**b**) staining of anti-IgE-treated Edox cells with an anti-FLAG antibody reactive with FLAG-tag on the Fcε expression cassette, showing the outcome of the treatment on ice or at 37 °C. Percentages refer to a decrease in fluorescent signal at 37 °C. MFI: mean fluorescence intensity.

**Figure 5 ijms-24-14997-f005:**
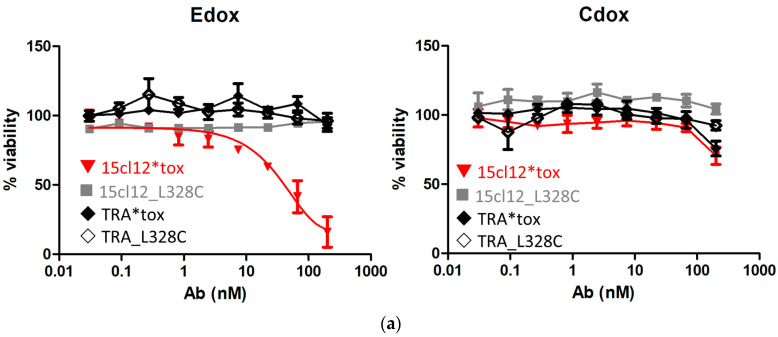
Cytotoxicity of the antibody–toxin conjugate. (**a**) Cytotoxicity of the antibodies to Edox and Cdox cell lines without additional crosslinker, trastuzumab (TRA), and its conjugate were applied as non-binding antibody controls; (**b**) cytotoxic effect exerted when the crosslinker was applied. 15cl12*tox: 15cl12_L328 coupled with toxin.

## Data Availability

The data presented in this study are available in this article and in the Appendix A.

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
