# Peer review of "Development of a Cytotoxic Antibody–Drug Conjugate Targeting Membrane Immunoglobulin E-Positive Cells"

_ijms, 2023, doi:10.3390/ijms241914997_

Round 1

Reviewer 1 Report

Development of cytotoxic Adc targeting Membrane immunoglobulin E positive cells

Recommendation: Publish after major revisions.

Comments:

The authors (Rodak et al) describe an interesting approach for ablating IgE-expressing cells using an antibody drug conjugate (ADC) approach. This idea is innovative and helps to push the concept of ADCs into therapeutic areas outside of oncology, which is a laudable goal.  Overall, the paper is logical, well written, and easy to read.  Experiments appear to have been conducted with scientific rigor.  However, there are several concerns that dampen my enthusiasm with this paper. I will outline the most significant concerns first, and then will list a number of minor concerns at the end.  Based on these concerns, I suggest a major revision is warranted – perhaps involving some additional experimentation if feasible – but at minimum requiring some additional discussion, explanation, and reformatting.

·       LCMS characterization of the conjugate (line 133 and Figure 1C).  The difference in the expected vs. theoretical mass of the conjugate differs by ~240 daltons.  The authors claim to have done ESI LCMS using a QTof instrument.  Typical error ranges for this type of instrument is +/- 3 daltons.  The massive difference between expected and actual mass indicates that a mistake was made in the use of the instrument or the conjugate itself is not the structure being proposed.  240 daltons is way outside of the acceptable error range.

·       Several of the graphs need to be replotted.  In particular Figures 2C and 5A.  The use of open circles and closed circles on the same line of the legend is highly confusing.  I had to re-read multiple times before I really understood what was being presented. One way to simplify the legend (and the graph) is to leave off the control (There is no reason to plot cells only and 2nd only in Fig 2C / and cells only in Fig 5A)

·       The internalization data (Fig 3) is not convincing to me.  While the authors are using a well-known method (AF488 quenching), they did not perform a time course and did not include a non-binding control antibody.  While I agree that the data suggests rapid internalization, the data does not convince me that the internalization is antigen mediated. Moreovoer, the NHS vs Cys conjugated is confusing. Are the authors suggesting that the NHS-coupled 15cl12 AF488 is internalized better than the Cys-coupled 15cl12 AF488?  If so, this needs to be explained. Performing these experiments using a non-binding control antibody would make the results much more clear and convincing.

·       The only functional assay presented that really demonstrates the mechanism of the ADC is the cytotoxicity experiment shown in Figure 5.  This seems to be rather weak data given that the title of the paper includes “development of cytotoxic antibody drug conjugate…”  The entire premise of the paper rests on the comparison of the MMAE conjugate in the engineered (Edox) cell line vs. the parental Edox cell line.  Figure 5B is pointless (maybe include in a supplemental figure) – as it is clear from figure 5A that the difference is statistically significant.  Typically, ADC papers demonstrate the mechanism of action for their ADC either by using an isotype control ADC or by co-treating with naked mAb to mitigate the cytotoxic effects.  As it currently stands, how do the authors know that the parental (Cdox) cell line is equally suseptable to MMAE toxicity?  Maybe the increased tox in the Edox cell line is simply due to increase sensitivity to the payload. Only an isotype control ADC would unravel this. (I agree this is unlikely, but given that the entire premise of the paper rests on this one figure, it seems important to me)

·       There is a lot of discussion about this crosslinker – but it wasn’t until I read the experimental section that it was stated that this is an goat anti-Fab.  Why exactly is this being employed? Are the authors suggesting that this would be part of the therapeutic approach?  Or are they just including it in order to artificially increase the cytotoxicity in order to make the statistical significance more compelling? 

Other minor concerns:

·       Site-specific coupling of the vcMMAE to the antibody requires a reduction/reoxidation process that should be cited. (line ~123)

·       Rather than “connecting the dots”, the figures should incorporate non-linear regression (curve fitting). [Fig 2C and 5A particularly]

·       Line 202, change “staining with quilizumab” to “binding of quilizumab”

·       The word “Compound” can be replaced with ADC, in the abstract

·       Qualizumab also binds to EMPD region of membranous IgE. Have the authors done a competitive assay to determine if Qualizumab can inhibit the binding ability 15cl12?

Author Response

  1. The authors (Rodak et al) describe an interesting approach for ablating IgE-expressing cells using an antibody drug conjugate (ADC) approach. This idea is innovative and helps to push the concept of ADCs into therapeutic areas outside of oncology, which is a laudable goal. Overall, the paper is logical, well written, and easy to read.  Experiments appear to have been conducted with scientific rigor.  However, there are several concerns that dampen my enthusiasm with this paper. I will outline the most significant concerns first, and then will list a number of minor concerns at the end.  Based on these concerns, I suggest a major revision is warranted – perhaps involving some additional experimentation if feasible – but at minimum requiring some additional discussion, explanation, and reformatting.

Answer 1: We thank the reviewer for positive comments, a very thorough revision and valuable remarks and ideas!

  1. LCMS characterization of the conjugate (line 133 and Figure 1C). The difference in the expected vs. theoretical mass of the conjugate differs by ~240 daltons. The authors claim to have done ESI LCMS using a QTof instrument.  Typical error ranges for this type of instrument is +/- 3 daltons.  The massive difference between expected and actual mass indicates that a mistake was made in the use of the instrument or the conjugate itself is not the structure being proposed.  240 daltons is way outside of the acceptable error range.

Answer 2: Thank you for the comment. We have performed the analysis in reducing conditions and indeed, the results show the molecular weight shift that fits one toxin per heavy chain (in duplicates, for independent protein and ADC preparations). The data is shown in Figure and commented in the Results section, and the information on the additional step is added to the Materials and Methods section. Also in our past experience, the MS measurements were more precise for the proteins around 50 kDa in size than for complete antibodies.

  1. Several of the graphs need to be replotted. In particular Figures 2C and 5A. The use of open circles and closed circles on the same line of the legend is highly confusing.  I had to re-read multiple times before I really understood what was being presented. One way to simplify the legend (and the graph) is to leave off the control (There is no reason to plot cells only and 2nd only in Fig 2C / and cells only in Fig 5A)

Answer 3: Thank you, we have modified the plots as you have proposed and split the lines of legend (only we still kept the controls).

  1. The internalization data (Fig 3) is not convincing to me. While the authors are using a well-known method (AF488 quenching), they did not perform a time course and did not include a non-binding control antibody. While I agree that the data suggests rapid internalization, the data does not convince me that the internalization is antigen mediated. Moreovoer, the NHS vs Cys conjugated is confusing. Are the authors suggesting that the NHS-coupled 15cl12 AF488 is internalized better than the Cys-coupled 15cl12 AF488?  If so, this needs to be explained. Performing these experiments using a non-binding control antibody would make the results much more clear and convincing.

Answer 4: Thank you for this valuable comment. We have performed the time course study of internalization and also included an irrelevant labelled non-binding antibody (cetuximab) labelled in the same way. We completely agree that presenting the results of internalization of the Cys-coupled antibody is only confusing at this point, so we removed it.

  1. The only functional assay presented that really demonstrates the mechanism of the ADC is the cytotoxicity experiment shown in Figure 5. This seems to be rather weak data given that the title of the paper includes “development of cytotoxic antibody drug conjugate…” The entire premise of the paper rests on the comparison of the MMAE conjugate in the engineered (Edox) cell line vs. the parental Edox cell line.  Figure 5B is pointless (maybe include in a supplemental figure) – as it is clear from figure 5A that the difference is statistically significant.  Typically, ADC papers demonstrate the mechanism of action for their ADC either by using an isotype control ADC or by co-treating with naked mAb to mitigate the cytotoxic effects.  As it currently stands, how do the authors know that the parental (Cdox) cell line is equally suseptable to MMAE toxicity?  Maybe the increased tox in the Edox cell line is simply due to increase sensitivity to the payload. Only an isotype control ADC would unravel this. (I agree this is unlikely, but given that the entire premise of the paper rests on this one figure, it seems important to me)

Answer 5: Thank you for the initiative, we have repeated the experiment with a non-binding conjugated antibody and there was no toxicity for the cell lines.

  1. There is a lot of discussion about this crosslinker – but it wasn’t until I read the experimental section that it was stated that this is an goat anti-Fab. Why exactly is this being employed? Are the authors suggesting that this would be part of the therapeutic approach? Or are they just including it in order to artificially increase the cytotoxicity in order to make the statistical significance more compelling?

Answer 6: Thank you for the comment. Indeed, the crosslinking of EMPD by antibody itself leads to an antiproliferative effect. While the toxin-coupled antibody was more cytotoxic to antigen-positive cells than the naked antibody, additional cross-link with the Fab enhanced this effect. We included this data to show that there might be more potential to the antibody as an ADC, perhaps with the introduction of additional IgE-binding sites (these comments are in the Discussion, fourth paragraph).

Other minor concerns:

  1. Site-specific coupling of the vcMMAE to the antibody requires a reduction/reoxidation process that should be cited. (line ~123)

Answer 7: Thank you, we have included the citation.

  1. Rather than “connecting the dots”, the figures should incorporate non-linear regression (curve fitting). [Fig 2C and 5A particularly]

Answer 8: Thank you, we have corrected all figures showing titrations so that they show non-linear regression. The “flat lines” (resulting from no response to control antibodies) in some Figures did not converge, and there we left connected dots.

  1. Line 202, change “staining with quilizumab” to “binding of quilizumab”

Answer 9: Thank you, corrected.

  1. The word “Compound” can be replaced with ADC, in the abstract

Answer 10: Thank you, corrected.

  1. Qualizumab also binds to EMPD region of membranous IgE. Have the authors done a competitive assay to determine if Qualizumab can inhibit the binding ability 15cl12?

Answer 11: Thank you, we have performed a competitive assay using quilizumab and a non-binding antibody (trastuzumab) as a control and found no competition (data presented in supplementary Figure 3). Comments are also included in the corresponding sections of Results and Materials and Methods.

Reviewer 2 Report

This paper is overall well-written on the subject of drug-conjugated antibody. I recommend publishing it after making some minor revisions.

1. I wish Figure 1a could be enlarged and its components indicated with arrows or similar marks.

2. Please verify whether the y-axis in Figure 1b is correct. The y-axis information in Figure 2a also needs to be verified.

3. It would be good to move the scale bar in Figure 3b to the lower right corner.

4. Is it possible to rephrase '2.5 Specific toxicity' and '4.5.1 Direct toxicity'?"

 Minor editing of English language required

Author Response

This paper is overall well-written on the subject of drug-conjugated antibody. I recommend publishing it after making some minor revisions.

We thank the reviewer for the careful reading of the manuscript, positive comments and the valuable remarks!

  1. I wish Figure 1a could be enlarged and its components indicated with arrows or similar marks.

Answer1: Thank you, we have enlarged and labeled the Figure elements with arrows.

  1. Please verify whether the y-axis in Figure 1b is correct. The y-axis information in Figure 2a also needs to be verified.

Answer 2: Thank you, the information on both axes is correct. We appreciate your concern and have added the explanation of abbreviation for the units to the legend.

  1. It would be good to move the scale bar in Figure 3b to the lower right corner.

Answer 3: Thank you, we have moved the error bar.

  1. Is it possible to rephrase '2.5 Specific toxicity' and '4.5.1 Direct toxicity'?"

Answer 4: Thank you, we have rephrased the titles.